# Retrieving Relevant EU Drone Legislation with Citation Analysis

**Gijs van Dijck** [1,2,*]🆔, **Alexandru-Daniel On** [3]🆔, **Jasper Snel** [2] **and Rohan Nanda** [1,4,5]

1.  Maastricht Law and Tech Lab., Maastricht University, 6211 LK Maastricht, The Netherlands; r.nanda@maastrichtuniversity.nl
2.  Brightlands Institute for Smart Society, Maastricht University, 6211 LK Maastricht, The Netherlands
3.  Maastricht European Private Law Institute, Maastricht University, 6211 LK Maastricht, The Netherlands; daniel.on@maastrichtuniversity.nl
4.  Institute of Data Science, Maastricht University, 6211 LK Maastricht, The Netherlands
5.  Department of Advanced Computing Sciences, Maastricht University, 6211 LK Maastricht, The Netherlands
*   Correspondence: gijs.vandijck@maastrichtuniversity.nl

**Abstract:** Can the retrieval of relevant unmanned aircraft systems (UAS) legislation be automated? In this article, references from and to EU legislation are used to create an overview that is subsequently compared to legislation considered relevant by subject-matter experts. The overlap between the results of the citation analysis and the expert overview is promising. Additionally, an approach was proposed and tested where, first, a relatively large number of laws were identified and, second, the laws that were considered relevant were selected. The findings reveal that this approach was successful at retrieving the majority of relevant laws. The results are relevant to researchers, policymakers, practitioners, and laypeople searching for relevant EU legislation on UAS.

**Keywords:** UAS; drones; legislation; automation; linked data; network analysis





## 1. Introduction

### 1.1. Research Problem

This study focuses on the findability of relevant legislation. In recent years, several European laws have been enacted or proposed on unmanned aircraft systems (UAS). The most prevalent, UAS-specific laws are regulations on unmanned aircraft systems and on third-country operators of unmanned aircraft systems [1], on the rules and procedures for the operation of unmanned aircraft [2], and on a regulatory framework for the U-space (i.e., an unmanned traffic management system) [3–5]. Other laws that can be applicable and relevant in addition to UAS-specific legislation can be categorized as aviation-specific and general-purpose legislation.

Previous research on UAS and UAS-related legislation has focused on analyzing and assessing the regulatory framework applicable to UAS operations [6–12], identifying the potential impact of UAS regulations [13], exploring various civil [14–17] and criminal liability [18,19] issues, including the liability of AI systems (which are and will be, more and more integrated into UAS technology) [20–22], exploring issues of privacy [23], cybersecurity [24] and data breaches [25], discussing the (potential) use of UAS in Law Enforcement and Capacity Monitoring [26], identifying processes and operational preventive measures to avoid malfunctioning and user negligence [27], and flagging potential gaps and uncertainties within the regulatory and liability frameworks that apply to UAS [28].

It can be difficult, even for those with legal expertise, to identify relevant UAS legislation. At the European level, the applicable rules are scattered across different pieces of regulations and directives. Moreover, which legislation can be considered relevant depends on the legal question one aims to answer. For instance, a UAS that causes personal injury due to a crash will likely lead to questions of liability for which liability law (or even criminal law) will need to be consulted. A UAS that flies too close to buildings and captures confidential work within the facility is likely to violate privacy laws, whereas a UAS that

closely flies over one's home may violate property law, more specifically, ownership rules that entail that property ownership includes not only the property itself but also a part of the airspace above the property. Considering that the applicable rules depend on the facts, a method for finding relevant UAS legislation should allow the user to select or focus on legislation that is applicable to the use case relevant to them.

This article discusses the automation of finding relevant UAS legislation. In this article, it is explored whether the finding of possible relevant drone legislation can be automated. Electronic legal information retrieval systems generally carry out the automatic identification of relevant legislation [29]. Prior research on legal information retrieval systems can be broadly classified into Boolean retrieval, Natural Language Processing (NLP)-based retrieval, and ontology-based retrieval. In Boolean information retrieval, documents are represented as sets of terms. A query is formulated in the form of a Boolean expression of terms. A document is retrieved if it satisfies the query expression. Many legal databases, such as Westlaw, Lexis, and Bloomberg Law, offer Boolean search functionality. Legal Boolean retrieval methods have been known to result in the retrieval of a large number of irrelevant documents and missing relevant documents that do not match the query expression [30]. In addition, they achieve a much lower recall (20%) compared to what manual (over 75% recall) legal researchers retrieve [31].

NLP-based retrieval methods utilize text similarity techniques for finding the most relevant legal documents for a user query. Text similarity techniques have been used in different legal information retrieval tasks, such as the identification of similar cases and judgments [32,33], identification of similar patents [34,35], identification of relevant legal statutes, and provisions for a user query [36,37]. The major text similarity techniques utilized in NLP-based retrieval include lexical, semantic, and language model-based representations. Text similarity techniques offer the advantage of ranking the retrieved legal documents on the basis of a similarity score in the order of relevance.

In ontology-based retrieval, external legal databases, thesauruses, vocabularies, or ontologies are utilized to incorporate semantic information for retrieving legal documents [38,39]. A major advantage of using an ontology-based information retrieval system is the availability of external semantic knowledge in a machine-readable form to enrich the retrieval process with the meaning of the terms. However, the user of such a system should be a domain expert or someone very familiar with the underlying ontology, semantics, and syntax of the query [40].

The present study will focus on reference-based retrieval, more specifically on whether the references in legislation to other laws can assist in identifying possibly relevant UAS legislation. It poses the question of whether the process of finding relevant UAS legislation can be automated by means of references from and to UAS legislation.

Any systematic search for relevant UAS legislation needs to grapple with several complicating factors that arise from the way the law is written, adopted, enforced, and organized in Europe. There are multiple layers of legislation that apply to UAS operations (Section 1.2), and different types of legislation can be found (Section 1.3).

### 1.2. Multi-Layered Legislation

The legislative and regulatory provisions applicable to UAS-related activities and operations can be organized according to their formal normative sources into four broad layers (or categories):

1. The International Layer: consisting primarily of international conventions, such as the Convention on International Civil Aviation (also known as Chicago Convention) [41] and its annexes, the Convention for the Unification of Certain Rules for International Carriage by Air (also known as the Montreal Convention) [42], and the non-binding yet highly influential global Standards and Recommended Practices (SARPs), Procedures, Model Regulations, and Guidance material for unmanned aviation developed by the International Civil Aviation Organization (ICAO) [43].

2.  The European Layer: consisting of European Union legislation, together with other international conventions (such as the European Convention on Human Rights), as well as norms derived from the case law of the Court of Justice of the European Union and the case law of the European Court of Human Rights.
3.  The National Layer: consisting of legislative acts and regulations adopted by individual Member States. National courts apply the legislative acts and regulations adopted by the Member states (and, in certain instances, also further develop the legal system).
4.  The Local Layer: consisting of legislative acts and regulations adopted by administrative subdivisions of the state, such as regions, provinces, or municipalities.

An example of multi-layered UAS legislation concerns the rules regarding the geographical zones in which UAS operations can or cannot take place. Article 15 of EU Regulation 2019/947 prescribes the possibility for a Member State to prohibit certain or all UAS operations when defining the UAS geographical zones [44]. A subsequent inspection of Dutch national law leads to Article 9 of the Flight Operations Decree ('Besluit vluchtuitvoering'), a general administrative order that states that geographical zones, as stated in Article 15 of EU Regulation 2019/947, can be defined or designated by ministerial orders [45]. Based on this above-mentioned provision, the ministerial order Zoning Regulations for Unmanned Aerial Vehicles ('Regeling zonering onbemande luchtvaartuigen') has been introduced. This order prohibits the operation of flights in the open category in specified geographical zones, which are visualized on a website [46]. The ministerial order sets rules in the following way: it first defines the specific type of area in which the flight is prohibited, and it then determines the specific coordinates surrounding those areas concerned. For example, based on the first paragraph of Article 2 'Regeling zonering onbemande luchtvaartuigen', it is prohibited to fly a UAS in the open category higher than 30 m in the geographical areas in which low-level flight by civil or military aircraft is allowed. By way of derogation from this first paragraph prescribing 30 m as the limit, there is an additional full prohibition in the second paragraph for flights of UAS subcategory A3 [47] in the areas defined in the article [48]. Then the article provides concrete coordinates to define which areas fall under these prohibitions. One of these areas is, for example, the 'GLV I (North Drenthe)' (the area enclosed by a line connecting the following positions: 53°03′45.00′′ N 006°43′30.35′′ E; 52°59′14.96′′ N 006°39′00.31′′ E; 3°01′00.20′′ N 006°38′00.54′′ E; 53°03′45.00′′ N 006°40′30.40′′ E; 53°03′45.00′′ N 006°43′30.35′′ E) [49]. Because explicit references between different legal orders (e.g., from EU law to Member State Law, or vice versa) are not readily available, this study focuses on UAS legislation in the EU.

*1.3. Types of EU Legislation*

In the most abstract terms, EU law can be divided into two basic categories: primary EU law and secondary EU law [50] (p. 53). Primary EU law consists of the founding treaties of the European Union [50] (p. 53), the most recent of which is the Treaty on the Functioning of the European Union (TFEU) [51]. Secondary EU law consists of legislation enacted by EU Institutions, case law (especially from the Court of Justice of the European Union), general principles of EU law, and international agreements entered into by the EU [50] (p. 53).

There are three types of binding legislative acts that EU law-making institutions can adopt: Regulations, Directives, and Decisions. In addition, EU institutions can adopt non-binding Recommendations or Opinions. Regulations are acts that are directly binding across the entirety of the European Union [52]. They do not require implementation in order to apply in each individual Member State [53]. Directives are legislative acts that, in principle, aim only to formulate the goals that European Union Member states should achieve in a certain area of policy [54]. Directives require implementation by the Member states, which means that the Member States must adopt national legislation to specify further how such goals are to be achieved [52]. Decisions are acts that are addressed to a determinate or determinable legal or natural person (such as a country, a company, a group of countries, or companies in a certain area of activity) and are binding only on the parties

to whom they are addressed [52]. Recommendations and Opinions allow EU institutions to formulate policy and statements in a non-binding way. They are used to suggest lines of action without imposing any obligations on the Member States or on private parties [52].

Of the four categories of legislation, the most relevant for the purpose of this article are Regulations and Directives because they are binding types of EU legislation. Both Regulations and Directives are of general applicability, but some Regulations and Directives have narrower domains than others. For instance, some regulations, such as the General Data Protection Regulation (GDPR) [55], are very general and create obligations for all data controllers and processors, whereas a Regulation such as 2019/947 [2] applies only to UAS operators and remote pilots. Whether a Directive or Regulation is adopted in any specific field depends on the type of competence the EU holds in that area, as well as the ever-evolving policy aims of the Union. Air traffic safety legislation is almost always adopted as a Regulation, whereas other domains, such as liability or the security of network and information systems, are regulated mainly through Directives. Transport is a shared competence between the EU and the Member States (see Article 4 of the TFEU), and air transport specifically can be regulated at the EU level according to Article 100(2) of the TFEU. This does not mean, however, that a search for UAS legislation can be limited to acts adopted on the basis of Article 100(2) of the TFEU. Some generally applicable legislative acts have a different legal basis, such as Article 114 of the TEFU (dealing with the harmonization of laws that have as their object the establishment and functioning of the internal market) or Article 16 of the TFEU (the right to protection of personal data).

For the purposes of this article, it is useful to further divide EU legislation into the following three categories: (1) UAS-specific; (2) aviation-specific; and (3) general-purpose legislation. UAS-specific legislation is legislation that was enacted with the explicit goal of regulating activities, products, and organizations connected with UAS technology. Aviation-specific legislation is more general, covering all types of aviation activities, products, and organizations involved in aviation. Because of its more general character, aviation-specific legislation applies to UAS as a subcategory of "aircraft", and it also compliments and influences UAS-specific legislation because of its more general character. Finally, general-purpose legislation is legislation that is not particularly designed for UAS or aircraft but can, nevertheless, be or become relevant to UASs. The application of general-purpose legislation to UASs and UAS activities can be said to be incidental because this kind of legislation was not written with UAS activities in mind. A great deal of general-purpose legislation even precedes the existence of the technologies that make UAS flights possible. Examples include the rules of property law, liability law, criminal law, or the laws that regulate cybersecurity and privacy. These rules create rights and obligations that apply to a variety of persons, goods, and services, which may also become relevant under the right circumstances to UAS activities. For instance, a liability rule that states that "a person who commits a wrongful act against another person, which can be attributed to him, is obliged to compensate the damage that the other person suffers as a result" [56] does not apply to most UAS operations because most UAS operations will go according to plan and will be perfectly safe. However, should a risk to life, bodily integrity, or property materialize (for instance, if a UAS collides with a person or a building), then general liability rules come into play, and any person who is at fault may have a duty to compensate the victim of the collision.

In the specific case of UASs, the normative and regulatory landscape is enriched but also made more complex because a large number of relevant norms are created by an agency of the European Union—the European Union Aviation Safety Agency (EASA). The EASA plays a key role in shaping EU secondary legislation by drafting and advising the European Commission on novel aviation-specific legislation and by proposing amendments during the legislative process. Additionally, the EASA adopts non-binding rules, such as the Acceptable Means of Compliance (AMC) and Guidance Material (GM), which provide much-needed guidance to all stakeholders involved in UAS operations. Finally,

the EASA also assists the national competent authorities in carrying out their legislative and regulatory tasks.

In the following sections, only hard law is analyzed; thus, the scope of the present study Recommendations and Opinions from EU law-making bodies, as well as Acceptable Means of Compliance, Guidance Material, and Opinions issued by the EASA, were excluded.

The remainder of the paper is organized as follows. Section 2 presents the materials and methods. The results are discussed in Section 3. Section 4 presents the discussion. The paper concludes in Section 5.

## 2. Materials and Methods

Our starting point was at the European level and, more specifically, the hard law of the European Union. From a practical perspective, this was an obvious choice because of the recent regulatory activity of the EU with regard to UAS operations [1,2], as well as the regulations regarding the U-space and U-space providers [3].

It was examined whether relevant UAS legislation, as defined by subject-matter experts, can be found or approximated by applying a linked data approach. Linked data is, in essence, a collection of best practices for publishing and connecting structured data such that it is machine-readable. It should have an explicit meaning and must be linked to external datasets, which are also linked to other external datasets on the Web. From a more technical point of view, linked data uses the Resource Description Framework (RDF) format to represent information on the Web. The advantage of using RDF is that it allows different data or things to connect on the Web using typed statements [57].

In the context of this paper, the references from one law to another as linked data were used. The European Commission stores these references as metadata in a repository (called CELLAR) and publishes them on EUR-Lex. By varying the degrees of separation, it was possible to control and vary the extent to which the references were removed from the source law, with the source law being the law (or laws) that were taken as a starting point. The CELEX-id of each source law—all laws have a unique identifier called CELEX—was taken, searched for in CELLAR for incoming and/or outgoing citations related to the particular CELEX-ID(s), and returned in the form of an edges list with CELEX-IDs that consisted of sources and targets, where the sources were the laws that were citing other laws and where the targets were the laws that were cited. Based on these edges lists, citation networks could be created.

It was possible to vary both the source depth and target depth. The source depth concerns the degree of separation regarding the references from other laws. A source depth of '1' means that the CELEX-IDs were retrieved from the laws that cite the source law. A source depth of '2' means that the CELEX-IDs returned cited the source law as well as the CELEX-IDs that cited the CELEX-IDs that cited the source law. This approach can be considered a forward-looking approach, given that generally and chronologically speaking, laws only cite laws from the past, not future laws (exceptions do exist, for instance, if two or more laws are drafted or enacted around the same time and, as a result, cite one another). The target depth is the degree of separation from the source law to other laws. For instance, a target depth of '2' means that the CELEX-IDs were returned from the laws that were cited in the source law as well as the laws that were cited in the laws that cited the source law. The target depth was backward-looking. Testing different combinations of source and target depths and comparing the results with the laws identified by the experts allowed us to calculate the overlap between the laws in the expert overview and the ones found by means of our linked data approach.

The strategy deployed was as follows. One or more starting laws (e.g., Regulation 2019/947) were selected, and references from and to this law were identified. Each analysis would start with a source depth and target depth of zero (meaning that no references to or from the source law are sought). After this, the source and target depth were incrementally increased by 1, in the sense that in every iteration, either the source depth or the target depth was increased (not both at the same time). The laws that were returned at each step

were compared to the laws mentioned by the experts. For this (1) the laws identified in the expert overview that were also found with our linked data approach (true positives), (2) the laws in the expert overview that were not found in our search (false negatives), and (3) the retrieved laws that were not listed in the expert overview (false positives) were selected. Based on this information, the precision (True positives/(True positives + False positives)) and recall (True positives/(True positives + False negatives)) were calculated, with precision indicating how many of the retrieved laws with the linked data approach were relevant, whereas recall was about how many of the relevant laws were retrieved. F1 scores were a tradeoff between precision and recall.

The relevant legislation was identified by consulting partners in a research project on UASs, including project partners with legal expertise in UAS legislation, comparative law, and practical expertise with UAS deployment (current or future). Representatives of nine project partners attended sessions in which an overview of relevant laws was created, of which two knowledge institutions and one UAS assessments and certification company provided the actual input. The corresponding CELEX number retrieved for each law mentioned in the overview was included in the analysis, meaning that the CELEX numbers in the expert overview were compared with the CELEX number in the searches that were conducted.

The scope of the expert overview was inherently arbitrary (see Section 1.3). Although we are confident that the vast majority of the UAS-specific legislation, such as the 2019/945 and 2019/947 Regulations [1,2], was identified by other experts, this is not necessarily the case for non-drone-specific legislation, such as laws on liability, cybersecurity, or trespassing. As explained above, what one would consider relevant for a UAS operation very much depends on the circumstances of the intended operation and on the legal questions that could or are expected to occur before, during, or after the operation is conducted.

The references were obtained through EUR-Lex, the official portal for providing access to European Union law. The information presented on EUR-Lex was retrieved from the common repository of metadata and content (CELLAR), which can be accessed through a SPARQL endpoint or through a RESTful API [58]. The available metadata includes the relationships between legislative documents. Outgoing and incoming references to or from documents can be retrieved one document at a time from either of these endpoints. This process can start from a distinct list of source documents. The resulting documents from this step can then be used as input for the next iteration of the retrieval. This process was repeated until the desired 'depth' of references was achieved.

With the references from and to the laws identified in our search, a network of laws was created where the nodes in the network consisted of the laws and where the citations form the edges that link the nodes. The application of network analysis allows us to automatically divide the laws into clusters, identify meaningful sub-topics from those clusters, and find central (relevant) laws within the clusters that can be distinguished from less central laws. The networks were analyzed and visualized in Gephi, an open-source software for the creation, manipulation, and visualization of networks [59].

The data collection was conducted on 13 June 2023. The data and code are publicly available at [anonymized].

## 3. Results

The laws that were identified by the experts vary in terms of the duty holder and what they aim to protect. They apply to UAS manufacturers, operators in the open, specific or certified categories of UAS operations, U-space service providers, all of the aforementioned, or to the Member States. The legislation that was considered relevant could be distinguished into six main categories: safety, more specifically, air safety, product safety, and safety investigations, cybersecurity, privacy, data governance, digital services, and liability. Table 1 provides an overview of the relevant legislation.

**Table 1.** Expert Overview of Relevant EU Legislation (M = UAS Manufacturers; O = UAS Operators in category open; S = UAS operators in a specific category; C = UAS Operators in a certified category; U = providers of U-space services; All = All of the aforementioned; MS = Member States). The overview builds on an initial overview provided by Filippo Tomasello (EuroUSC).

| Domain | Applicable to | Reference |
|---|---|---|
| Safety | All | 2018/1139 [60] |
| | MS | 996/2010 [61] |
| | None | 1025/2012 [62] |
| | M | 765/2008 [63] |
| | | 768/2008 [64] |
| | | 2006/42 [65] |
| | | 2009/48 [66] |
| | | 748/2012 [67] |
| | | 2015/640 [68] |
| | | 2019/945 [1] |
| | All | 300/2008 [69] |
| | | 376/2014 [70] |
| | O, S | 2019/947 [2] |
| | S, C | 923/2012 [71] |
| | | 2021/666 [5] |
| | C | 1178/2011 [72] |
| | | 965/2012 [73] |
| | | 1321/2014 [74] |
| | | 1332/2011 [75] |
| | | 2018/1048 [76] |
| | U | 2017/373 [77] |
| | | 2021/664 [3] |
| | | 2021/665 [4] |
| Cybersecurity | All | 2016/1148 [78] |
| | All | 2019/881 [79] |
| Privacy | All | 2016/679 [55] |
| Data Governance | All | 2022/868 [80] |
| Digital Services | All | 2022/2065 [81] |
| Digital Identity | All | European Digital Identity (reform proposal) [82] |
| Liability and Insurance | M | 85/374 [83] |
| | O, S, C | 785/2004 [84] |

One of the laws mentioned in the expert overview concerned, at the time of writing this paper, a proposal for reform. It was decided not to include the proposal in our analysis. This left us with 30 laws identified by the experts.

*Citation Analysis*

As explained above, the citation analysis consisted of the following:

1. The selection of one or more laws. These laws served as a starting point for the search (hereafter: source law(s)).
2. The collection of citations with different source and target depths, starting with zero (i.e., no references to or from the source law) and incrementally increasing the depth by 1.
3. A comparison of the results of the citation analysis (referred to as 'search results') to the laws in the expert overview. Precision, recall, and F1 scores were used to evaluate the findings.

We were first interested in the recall: do the laws in the expert overview appear in the search results? A high recall would suggest that the citation analysis was able to reproduce the results by the experts. The results reveal that a recall could be achieved that approximates 0.70 (Tables 2–5) with just a few steps (degree depth), meaning that around 70% of the laws mentioned in the expert overview could be retrieved.

**Table 2.** Precision, Recall, and F1 scores for different source and target depths, with Regulation 2021/664 and 2019/945 as source laws.

| Target depth | 0 | 0 | 0 | 0 | 1 | 1 | 1 | 1 | 1 | 2 | 2 | 2 | 2 | 2 | 3 | 3 | 3 | 3 | 3 | 4 | 4 | 4 | 4 | 4 |
|---|---|---|---|---|---|---|---|---|---|---|---|---|---|---|---|---|---|---|---|---|---|---|---|---|
| Source depth | 1 | 2 | 3 | 4 | 0 | 1 | 2 | 3 | 4 | 0 | 1 | 2 | 3 | 4 | 0 | 1 | 2 | 3 | 4 | 0 | 1 | 2 | 3 | 4 |
| Precision | 0.50 | 0.44 | 0.36 | 0.08 | 0.76 | 0.64 | 0.62 | 0.57 | 0.25 | 0.21 | 0.21 | 0.20 | 0.20 | 0.15 | 0.07 | 0.07 | 0.07 | 0.07 | 0.06 | 0.03 | 0.03 | 0.03 | 0.03 | 0.03 |
| Recall | 0.13 | 0.13 | 0.13 | 0.13 | 0.53 | 0.53 | 0.53 | 0.53 | 0.53 | 0.70 | 0.70 | 0.70 | 0.70 | 0.70 | 0.70 | 0.70 | 0.70 | 0.70 | 0.70 | 0.70 | 0.70 | 0.70 | 0.70 | 0.70 |
| F1 | 0.21 | 0.21 | 0.20 | 0.10 | 0.63 | 0.58 | 0.55 | 0.55 | 0.34 | 0.33 | 0.32 | 0.32 | 0.31 | 0.31 | 0.13 | 0.12 | 0.12 | 0.12 | 0.11 | 0.06 | 0.06 | 0.06 | 0.06 | 0.05 |

**Table 3.** Precision, Recall, and F1 scores for different source and target depths, with Regulation 2021/664 as source law.

| Target depth | 0 | 0 | 0 | 0 | 1 | 1 | 1 | 1 | 1 | 2 | 2 | 2 | 2 | 2 | 3 | 3 | 3 | 3 | 3 | 4 | 4 | 4 | 4 | 4 |
|---|---|---|---|---|---|---|---|---|---|---|---|---|---|---|---|---|---|---|---|---|---|---|---|---|
| Source depth | 1 | 2 | 3 | 4 | 0 | 1 | 2 | 3 | 4 | 0 | 1 | 2 | 3 | 4 | 0 | 1 | 2 | 3 | 4 | 0 | 1 | 2 | 3 | 4 |
| Precision | 0.67 | 0.67 | 0.67 | 0.67 | 1.00 | 0.88 | 0.88 | 0.88 | 0.88 | 0.71 | 0.69 | 0.69 | 0.69 | 0.69 | 0.18 | 0.18 | 0.18 | 0.18 | 0.18 | 0.06 | 0.06 | 0.06 | 0.06 | 0.06 |
| Recall | 0.07 | 0.07 | 0.07 | 0.07 | 0.23 | 0.23 | 0.23 | 0.23 | 0.23 | 0.67 | 0.67 | 0.67 | 0.67 | 0.67 | 0.70 | 0.70 | 0.70 | 0.70 | 0.70 | 0.70 | 0.70 | 0.70 | 0.70 | 0.70 |
| F1 | 0.12 | 0.12 | 0.12 | 0.12 | 0.38 | 0.38 | 0.38 | 0.38 | 0.38 | 0.69 | 0.68 | 0.68 | 0.68 | 0.68 | 0.29 | 0.29 | 0.29 | 0.29 | 0.29 | 0.11 | 0.11 | 0.11 | 0.11 | 0.11 |

**Table 4.** Precision, Recall, and F1 scores for different source and target depths, with Regulation 2021/665 as source law.

| Target depth | 0 | 0 | 0 | 0 | 1 | 1 | 1 | 1 | 1 | 2 | 2 | 2 | 2 | 2 | 3 | 3 | 3 | 3 | 3 | 4 | 4 | 4 | 4 | 4 |
|---|---|---|---|---|---|---|---|---|---|---|---|---|---|---|---|---|---|---|---|---|---|---|---|---|
| Source depth | 1 | 2 | 3 | 4 | 0 | 1 | 2 | 3 | 4 | 0 | 1 | 2 | 3 | 4 | 0 | 1 | 2 | 3 | 4 | 0 | 1 | 2 | 3 | 4 |
| Precision | 1.00 | 0.67 | 0.67 | 0.67 | 1.00 | 1.00 | 0.67 | 0.67 | 0.67 | 1.00 | 1.00 | 0.88 | 0.88 | 0.88 | 0.71 | 0.71 | 0.69 | 0.69 | 0.69 | 0.18 | 0.18 | 0.18 | 0.18 | 0.18 |
| Recall | 0.07 | 0.07 | 0.07 | 0.07 | 0.07 | 0.07 | 0.07 | 0.07 | 0.07 | 0.23 | 0.23 | 0.23 | 0.23 | 0.23 | 0.67 | 0.67 | 0.67 | 0.67 | 0.67 | 0.70 | 0.70 | 0.70 | 0.70 | 0.70 |
| F1 | 0.12 | 0.12 | 0.12 | 0.12 | 0.12 | 0.12 | 0.12 | 0.12 | 0.12 | 0.38 | 0.38 | 0.37 | 0.37 | 0.37 | 0.69 | 0.69 | 0.68 | 0.68 | 0.68 | 0.29 | 0.29 | 0.29 | 0.29 | 0.29 |

**Table 5.** Precision, Recall, and F1 scores for different source and target depths, with Regulation 2019/947 as source law.

| Target depth | 0 | 0 | 0 | 0 | 1 | 1 | 1 | 1 | 1 | 2 | 2 | 2 | 2 | 2 | 3 | 3 | 3 | 3 | 3 | 4 | 4 | 4 | 4 | 4 |
|---|---|---|---|---|---|---|---|---|---|---|---|---|---|---|---|---|---|---|---|---|---|---|---|---|
| Source depth | 1 | 2 | 3 | 4 | 0 | 1 | 2 | 3 | 4 | 0 | 1 | 2 | 3 | 4 | 0 | 1 | 2 | 3 | 4 | 0 | 1 | 2 | 3 | 4 |
| Precision | 0.60 | 0.44 | 0.24 | 0.04 | 1.00 | 0.83 | 0.69 | 0.46 | 0.10 | 0.32 | 0.32 | 0.32 | 0.28 | 0.12 | 0.07 | 0.07 | 0.07 | 0.07 | 0.05 | 0.03 | 0.03 | 0.03 | 0.03 | 0.02 |
| Recall | 0.10 | 0.13 | 0.13 | 0.13 | 0.30 | 0.33 | 0.37 | 0.37 | 0.37 | 0.53 | 0.57 | 0.60 | 0.60 | 0.60 | 0.53 | 0.57 | 0.60 | 0.60 | 0.60 | 0.57 | 0.57 | 0.60 | 0.60 | 0.60 |
| F1 | 0.17 | 0.21 | 0.17 | 0.06 | 0.46 | 0.48 | 0.48 | 0.41 | 0.16 | 0.40 | 0.41 | 0.41 | 0.38 | 0.20 | 0.12 | 0.13 | 0.13 | 0.13 | 0.10 | 0.05 | 0.05 | 0.05 | 0.05 | 0.05 |

The focus on recall is likely to lead to an overinclusion of laws that are not relevant or not highly relevant. This is not necessarily problematic, particularly when those laws can be clustered, as it would allow for ignoring one or more clusters of laws. Network analysis, community detection in particular, makes it possible to cluster laws. One strategy could, therefore, be first to select all possibly relevant laws (high recall) and to narrow down the selection (increase precision) subsequently. This strategy fits the notion that the legislation that can be considered relevant depends on the legal question one aims to answer, which in turn, depends on the circumstances at hand.

Increasing the depth (source and/or target depth) was the most likely strategy to result in the highest possible recall. To explore whether a visualized network could be reasonably interpreted when it includes a large number of laws that appear in the expert overview (high recall) and possibly a large number of laws that appear in the search results but not in the expert overview (low precision), the combination of source and target depth was

selected that yielded the highest precision given the highest recall. The best results were obtained when selecting Regulations 2019/945 and 2021/664 as source laws, with source depth = 0 and target depth = 2. With these parameters, a recall of 0.70 and a precision of 0.21 was obtained (F1 = 0.33) (Table 2).

The network is visualized below in Figure 1.

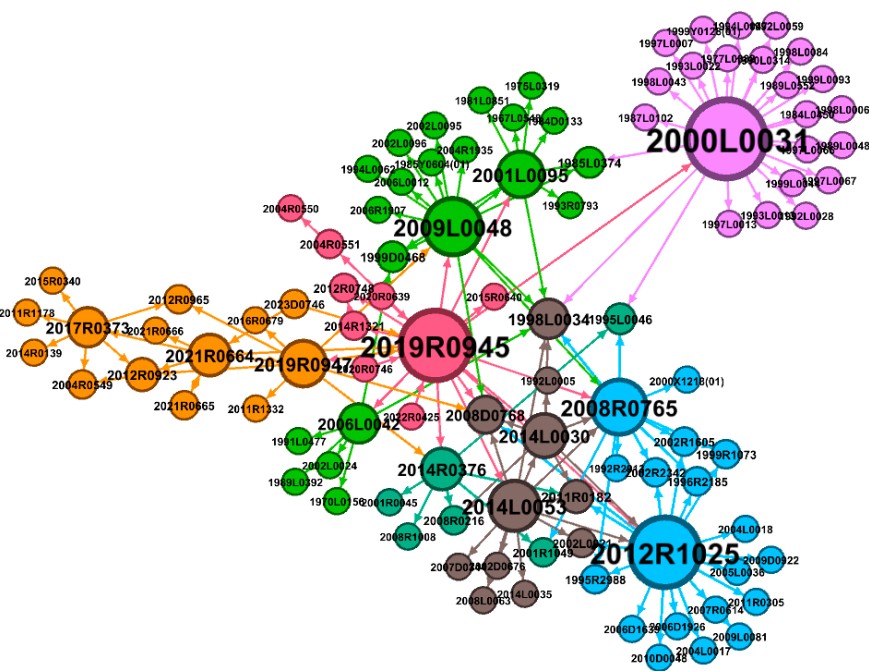

**Figure 1.** Network visualization (source laws = Regulation 2021/664, Regulation 2019/945, target depth = 2, source depth = 0). Notes: Communities are represented by different colors and based on the Louvain method for community detection. Node size is adjusted for ingoing and outgoing references combined. The labels are formatted as follows: year, type of legislation (R = Regulation, L = Directive, D = Decision), and legislation ID. The visualization can be inspected in more detail by downloading the Gephi file in the repository that is made publicly available at https://doi.org/10.34894/HLWZLL.

What is noticeable in this network is that the UAS-specific legislation was clustered in two communities (in red and orange), whereas the central laws in the other clusters concern subjects like electronic commerce (in purple), toy and product safety (in green), electromagnetic compatibility and radio equipment (in brown), actions following incidents in civil aviation (in dark green/turquoise), and standardization, accreditation, and market surveillance (in blue).

The network contains 21 laws out of the 30 that also appear in the expert overview. Compared to the previous results, the first network in particular, the search yielded one additional law that also appeared in the expert overview: 1985L0374 (on product liability). Because the precision is of secondary relevance, the network also includes 77 laws that are not in the expert overview that may turn out to be relevant, depending on the legal question that is at stake.

A high recall alone is not necessarily informative. It could mean that the search results include many relevant laws, but the results may also include many irrelevant laws. A high proportion of irrelevant laws can lead to relevant results that remain hidden in a pile of irrelevant ones. We, consequently, given a high recall (close to 0.70), sought source nodes that achieved higher precision while losing minimal recall. We will call this 'efficient retrieval' moving forward.

To obtain efficient retrieval, we first tried to simplify the search by minimizing the depth (source depth, target depth). The highest F1 score (0.63) was obtained when selecting a target depth of 1 and a source depth of 0 (Figure 2), but then the recall is lower (0.53) than when selecting Regulation 2021/664 as the sole starting point (recall = 0.67).

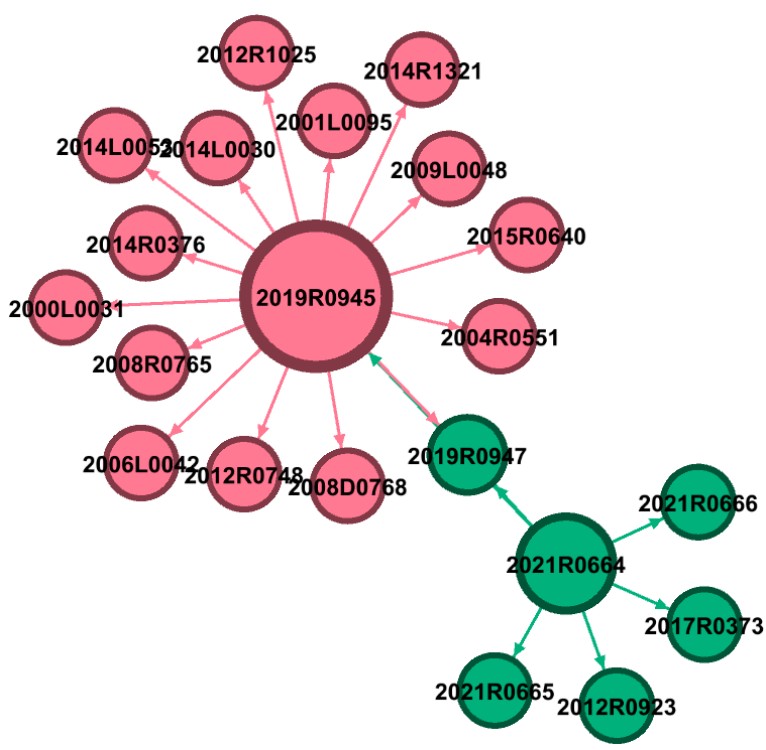

**Figure 2.** Network visualization (source laws = Regulation 2021/664, Regulation 2019/945, target depth = 1, source depth = 0). Notes: Communities are represented by different colors and based on the Louvain method for community detection. Node size is adjusted for ingoing and outgoing references combined. The labels are formatted as follows: year, type of legislation (R = Regulation, L = Directive, D = Decision), and legislation ID. The visualization can be inspected in more detail by downloading the Gephi file in the repository that is made publicly available at https://doi.org/10.34894/HLWZLL.

A more efficient retrieval was obtained when selecting Regulation 2021/664 [3] or Regulation 2021/665 [4] as the source law. Both will regulate the so-called U-space, a set of services and procedures related to the access to airspace for UASs. A recall of 0.67, a precision of 0.71, and an F1 score of 0.69 were obtained with target depth = 2 and source depth = 0, thereby selecting the references in Regulation 2021/664, as well as the references mentioned in the references (Table 3).The findings entail that 67% of the laws in the expert overview could be retrieved and that 71% of the laws in the search results also appear in the expert overview and appear in the search results. Increasing the target depth resulted in only a few more laws that also appear in the expert overview (recall increases from 0.67 to 0.70) and in a substantially larger number of laws not mentioned by the experts (precision dropped from 0.71 to 0.18) (Table 3). An increase in the source depth did not impact the results, which can be explained by the fact that Regulation 2021/664 is a recent piece of legislation that has simply not yet been cited often.

The laws missing in the search results concern:

- Regulation 2018/1139 (on civil aviation safety);
- Regulation 2010/996 (on accident and incident investigation and prevention);
- Regulation 2008/300 (on civil aviation security);
- Regulation 2018/1048 (on airspace usage);
- Directive 2016/1148 (on the security of network and information systems);
- Regulation 2019/881 (Cybersecurity Act);
- Directive 374/1985 (on product liability);
- Regulation 2004/785 (on insurance requirements for air carriers and aircraft operators);
- Regulation (EU) 2022/868 (Data Governance Act);
- Regulation (EU) 2022/2065 (Digital Services Act).

In turn, the legislation that was found in the search results but that did not appear in the expert overview concerned:

- Regulation 139/2014 (on aerodromes) [85];
- Regulation 549/2004 (on laying out a framework for the harmonization of European air traffic control) [86];
- Regulation 551/2004 (on the organization and use of European airspace) [87];
- Regulation 2015/340 (on air traffic controllers' licenses and certificates) [88];
- Directive 2014/53 (on the making available on the market of radio equipment) [89];
- Directive 2001/95 (on product safety) [90];
- Directive 2000/31 (on electronic commerce) [91];
- Directive 2014/30 (on electromagnetic compatibility) [92].

Various laws in the search results that did not appear in the expert overview can arguably be relevant in the context of UASs. Airspace and air traffic control legislation applies to and can affect UAS operations, whereas laws on product safety, radio equipment, and electromagnetic compatibility can be relevant for assessing the features and liability of UASs.

Here, we observe four communities when visualizing the network (Figure 3). One community (in blue) consists of legislation on U-space, one (in green) on air traffic related-matters, one (in purple) of laws mostly related to the specifications UASs are subjected to, and one community (in orange) on the laws regarding the carrying out of the operation, but also on data protection (General Data Protection Regulation—GDPR) and laws that could be brought under one of the other clusters.

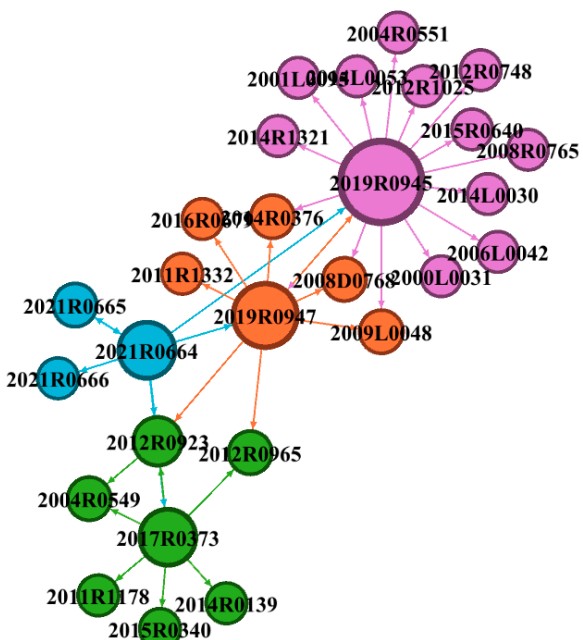

**Figure 3.** Network visualization (source law = Regulation 2021/664, target depth = 2, source depth = 0). Notes: Communities are represented by different colors and based on the Louvain method for community detection. Node size is adjusted for ingoing and outgoing references combined. The labels are formatted as follows: year, type of legislation (R = Regulation, L = Directive, D = Decision), and legislation ID. The visualization can be inspected in more detail by downloading the Gephi file in the repository that is made publicly available at https://doi.org/10.34894/HLWZLL.

The results are similar when selecting Regulation 2021/665 as the source law (Table 4). The only difference is that efficient retrieval was found with a target depth of 3 instead of 2 (F1 = 0.69, precision = 0.71, and recall = 0.67). The reason for this is that Regulation 2021/665 only cites Regulation 2021/664—one additional degree of separation.

The overlap between the expert overview and the search results decreased when other UAS-specific legislation was selected as a starting point (Table 5). For instance, when selecting Regulation 2019/947 [2], lower recall and precision scores were obtained. Depending on the source and target depths that were selected, either fairly high recall or precision scores could be obtained. For instance, if a source and target depth of 2 was selected, the recall was 0.60, and the precision was 0.32. A source depth and target depth of 1 increased the precision to 0.83 (more of the search results also appeared in the expert overview) but lowered the recall to 0.33 (fewer of the laws in the expert overview were retrieved).

The selection of Regulation 2019/945 [1] as the source law resulted in a maximum recall score of 0.60 and in a maximum precision score of 0.69 (Table 6). Also, here, there are no instances where both the recall and the precision were relatively high, as illustrated by the low F1 scores.

**Table 6.** Precision, Recall, and F1 scores for different source and target depths, with Regulation 2019/945 as source law.

| Target depth | 0 | 0 | 0 | 0 | 1 | 1 | 1 | 1 | 1 | 2 | 2 | 2 | 2 | 2 | 3 | 3 | 3 | 3 | 3 | 4 | 4 | 4 | 4 | 4 |
|---|---|---|---|---|---|---|---|---|---|---|---|---|---|---|---|---|---|---|---|---|---|---|---|---|
| Source depth | 1 | 2 | 3 | 4 | 0 | 1 | 2 | 3 | 4 | 0 | 1 | 2 | 3 | 4 | 0 | 1 | 2 | 3 | 4 | 0 | 1 | 2 | 3 | 4 |
| Precision | 0.43 | 0.44 | 0.36 | 0.08 | 0.69 | 0.57 | 0.57 | 0.52 | 0.21 | 0.18 | 0.18 | 0.19 | 0.18 | 0.13 | 0.05 | 0.06 | 0.06 | 0.06 | 0.05 | 0.02 | 0.02 | 0.03 | 0.03 | 0.03 |
| Recall | 0.10 | 0.13 | 0.13 | 0.13 | 0.37 | 0.40 | 0.43 | 0.43 | 0.43 | 0.53 | 0.53 | 0.60 | 0.60 | 0.60 | 0.53 | 0.57 | 0.60 | 0.60 | 0.60 | 0.53 | 0.57 | 0.60 | 0.60 | 0.60 |
| F1 | 0.16 | 0.21 | 0.20 | 0.10 | 0.48 | 0.47 | 0.49 | 0.47 | 0.47 | 0.27 | 0.27 | 0.28 | 0.28 | 0.22 | 0.10 | 0.10 | 0.11 | 0.11 | 0.10 | 0.04 | 0.05 | 0.05 | 0.05 | 0.05 |

The search with the highest F1 score resulted in an ego network of the regulation, along with a cluster that groups together the rules on the operation and the airspace or U-space (Figure 4).

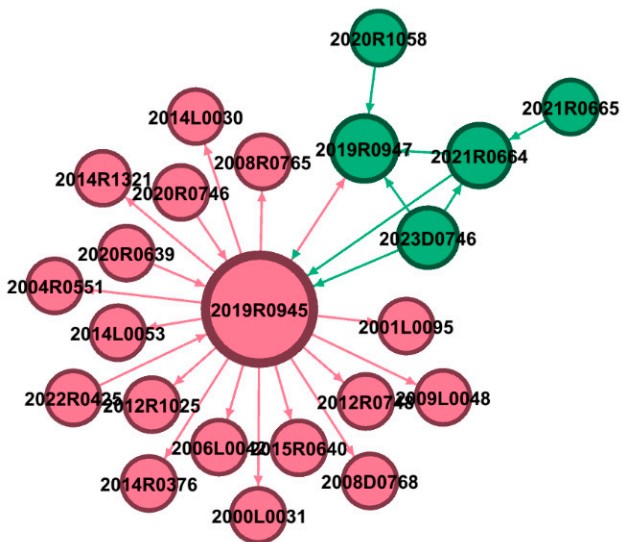

**Figure 4.** Network visualization (source law = Regulation 2019/945, target depth = 1, source depth = 2). Notes: Communities are represented by different colors and based on the Louvain method for community detection. Node size is adjusted for ingoing and outgoing references combined. The labels are formatted as follows: year, type of legislation (R = Regulation, L = Directive, D = Decision), and legislation ID. The visualization can be inspected in more detail by downloading the Gephi file in the repository that is made publicly available at https://doi.org/10.34894/HLWZLL.

Another strategy to increase the precision without losing substantial recall is to filter the network based on the most central laws. Here, we filtered the most central laws using the concept of degree: the combined number of incoming and outgoing references from

or to other legislation (multiple citations from the same law to another (same) law counts as one citation). The idea of filtering by degree is that it is preferred to exclude laws that are rarely referenced or are referencing other laws. Ideally, the most central laws in the network coincide with the laws that appear in the expert overview.

As a first iteration, the laws with at least two incoming and/or outgoing references were selected (Degree > 2—note that all nodes in the other networks had a degree of 1). The results reveal that the overlap between the laws in the filtered network and the ones in the expert overview proved to be limited. Although the precision increased substantially, from 0.21 to 0.57, this came at a loss of recall (from 0.75 to 0.52). These results are substantially worse than the findings obtained in previous networks, the first network in particular, where the precision was 0.70 and the recall 0.68.

## 4. Discussion

This study focused on the automation of finding relevant unmanned aircraft systems (UAS) legislation through the use of linked data and network analysis. The overlap that was found between the expert overview and the linked data approach suggests that EU legislation relevant to UAS can, for its majority, be found by means of an automated approach that leverages the availability of linked data as metadata. At the same time, not all laws deemed relevant by the experts could be retrieved, meaning that a linked data and network approach is likely not to be complete. This particularly applies to aviation-specific and general-purpose legislation. In contrast, all of the UAS-specific legislation could be retrieved in the searches performed in this study.

The retrieval would have been more successful if the source and target depth could be determined for each individual node (e.g., source depth = 1 for node X, source depth = 3 for node Y, etc.). This approach, however, has limited practical relevance, as an expert overview is required for a proper evaluation, which is often lacking in practice. The results may also have been better if the experts had different 'use cases' (circumstances) in mind. The laws listed in the expert overview suggest that the experts had drone operations in mind when selecting the laws, along with regulation on data (Data Governance Act, GDPR, and Cybersecurity) and digital services (Digital Services Act), although the applicability of the latter to UASs is debatable. Because an almost infinite number of situations can occur in reality, each of which may affect the laws that are applicable to the particular situation, and because experts may not always be readily available or affordable, a citation approach can be helpful or complimentary for identifying relevant or applicable legislation, perhaps even for experts. What makes a citation analysis promising is that it empowers users to construct networks iteratively and to zoom in on sub-networks, thereby allowing users to select the results relevant to their respective use cases.

Using citation data is not the only possibility for identifying relevant legislation. Other data-science-based or AI-based approaches exist for identifying relevant legislation in an automated manner. One option is to develop an ontology that maps the relationships between laws, provisions within laws, or relationships between concepts mentioned in legal provisions. Developing an ontology, however, requires substantial human input, whereas ontology development in an automated way is an avenue that needs further exploration and experimentation. One may also search for semantically similar laws or provisions given a selected input text. Semantic similarity may not be a promising approach since it often puts emphasis on word frequency, whereas frequently used terms are unlikely to correlate with the importance of the legal topic at hand. Previous research on court cases showed that, although there is a substantial overlap, the similarity of the full texts of court decisions does not closely mirror citation behavior [32].

Another limitation of this study is that it focuses on legislation at the EU level. Applicable and relevant UAS legislation is found on the European level as well as on the national level and in standards. One could conduct a similar exercise as performed in this article in order to find applicable legislation on the Member State level or in standards. However, this does require (1) explicit references from national legislation and/or standards to EU

legislation (or vice versa) and (2) the automatic detection of the references (preferably available as metadata). For instance, references in Member State law to Regulation 2019/947 are often not recognized by the reference detection software currently in place (if such software exists), which complicates the automated detection of references, in this case, to EU legislation on UASs. Moreover, reference detection software does not exist or is not applied in several countries, making automatic detection based on citation (meta)data not realistic.

Finally, mapping the norms applicable to UASs requires us to go beyond hard law. Standards complement and further specify hard law norms. Sometimes, EU legislation or national legislation may refer to a standard or a set of standards, thus integrating the technical norms from the referred-to-standard into the system of laws, consequently turning said technical standards into something that can perhaps be called *quasi-hard-law*. Examples of references to standards in EU legislation can be found in Article 14 of Regulation 2019/947, UAS.LUC.030 para. 2(a) of Regulation 2019/947, as well as Articles 6(4), 12, 13(2)(a), 22(7)(c), 23, 30(3), 36(5)(b), 37(3), and 40(4) of Regulation 2019/945. Most of the above-mentioned references are generic and do not explicitly identify the relevant standards (a noteworthy exception is Article 40(4) of Regulation 2019/945, which refers explicitly to standard ANSI/CTA-2063-A-2019, Small Unmanned Aerial Systems Serial Numbers, 2019), thus making the European norm more flexible and adaptable to technological innovation. At the same time, this complicates the automated identification of the most relevant standards. The inclusion of standards in the automated identification of UAS rules is an avenue for future research.

## 5. Conclusions

This study explored whether references from and to EU laws could be leveraged to identify relevant EU legislation on UASs. To analyze this, the results of a linked data and network analysis approach were compared to the laws identified by subject-matter experts, which were used as ground truth. The overlap was calculated, that is, the laws identified by the experts compared to the laws found through the citation analysis. The comparison of findings relevant to UAS legislation on the EU level by consulting experts and based on references from and to legislation showed a substantial overlap between the two approaches.

The findings suggest that various strategies may be adopted when searching for relevant legislation by means of citation analysis. One strategy is to take recent legislation as a starting point and identify potentially relevant laws by searching for references in (and perhaps to) the source law. This approach is likely to strike a balance between precision (the proportion of the search results that is relevant) and recall (the proportion of potentially relevant laws that is in the search results). Another strategy is to maximize the recall by starting with one or more laws and incrementally increase the target and/or source depth until, for the user, an acceptable network is created. The user can subsequently select the groups of laws (communities) of interest, create a subnetwork of the laws within those communities, and further inspect the laws within their respective communities. Because the laws that are relevant depend on the circumstances or the legal question at hand, the latter approach might prove useful for both legal experts and those who have some but lack in-depth legal expertise.

The results and approaches that were explored proved promising for retrieving relevant UAS legislation. Future research could take up the challenge of creating citation networks on multiple levels, for example, the EU level and the national (Member State) level. With a linked data approach, it will then become possible to connect laws on one level (e.g., EU) to all legislation on the national level. It is also possible to include other sources, such as standards, yet this requires the legislation in standards to be published in a linked data format. Finally, a citation analysis like the one presented in this article may be connected to the work that parses legislative provisions into obligations, prohibitions, permissions, IF-THEN statements, etc. This way, it will become possible to, given a set of

relevant laws, extract, for instance, all obligations that apply to an operator and extract the conditions that are related to the obligation. Such a combination of approaches has the potential to benefit automated compliance checking, whether it concerns UASs or other phenomena.

**Author Contributions:** Conceptualization, G.v.D.; methodology, G.v.D.; software, J.S.; validation, G.v.D.; formal analysis, G.v.D. and A.-D.O.; investigation, G.v.D.; resources, R.N., A.-D.O. and G.v.D.; data curation, G.v.D.; writing—original draft preparation, G.v.D., A.-D.O. and R.N.; writing—review and editing, G.v.D., A.-D.O. and R.N.; visualization, G.v.D.; supervision, G.v.D.; funding acquisition, G.v.D. All authors have read and agreed to the published version of the manuscript.

**Funding:** This research was funded by the European Union H2020 Research and Innovation Program under Grant Agreement No. 101006828—Flying Forward 2020.

**Data Availability Statement:** The data and code are available at https://doi.org/10.34894/HLWZLL.

**Acknowledgments:** We thank Filippo Tomasello of EuroUSC and other project partners for their valuable input for the expert overview.

**Conflicts of Interest:** The authors declare no conflict of interest. The funders had no role in the design of the study; in the collection, analyses, or interpretation of data; in the writing of the manuscript, or in the decision to publish the results.

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
