# Peer review of "Retrieving Relevant EU Drone Legislation with Citation Analysis"

_drones, doi:10.3390/drones7080490_

Round 1
Reviewer 1 Report
In the reviewed paper, the Authors presented the results of retrieving relevant EU drone legislation with citation analysis. In the paper, references from and to EU legislation are used to create an overview that is subsequently compared to legislation considered relevant by subject-matter experts. This is a review paper. In my opinion, the paper can be considered for publication, after taking into account the following remarks:
- paper doesn't meet the paper template requirements. The paper should be formatted into Drones paper template,
- before the paper publishing, the English should be carefully checked by a professional Native Speaker,
- at the end of the Introduction section, the Authors should write what was contained in each paper section,
- footnotes should be replaced with classic references to literature like [x], [y], [z], ...,
- the meaning of some of the used acronyms is not explained in the paper text. It should be improved,
- in the section called "2. Materials and Methods" some detailed information about gathered sources should be provided,
- information/captions provided on the figure 1 in majority cases are unreadable and should be improved. This remark is dedicated to all similar cases in the paper text,
- the Conclusions section is very short, is written in a very general way, and should be extended by adding some detailed conclusions from presented in the paper review,
- there are only 15 reference items in the review paper?
Thank you very much.
Before the paper publishing, the English should be carefully checked by a professional Native Speaker. Some of the sentences are written in such a way, that they are difficult for understanding,
-
Author Response
We thank the reviewer for the comments and suggestions. Find attached our responses.

Reviewer 2 Report
Very interesting topic and great job!
Here are a few comments/questions:
1. It is not very clear how relativity of two regulations are decided. Can you explain a little more how source depth and target depth are established?
2. How are expert selected and how are their opinions quantified?
3. Why two communities Red and Orange in Fig 1?
Author Response

(The authors gave the same response as above.)

Reviewer 3 Report
Very good idea for a legal article. Very interesting analysis.
I would just only add more available literature on the regulations on drones.
Author Response

(The authors gave the same response as above.)

Reviewer 4 Report
My area of expertise is on drone law, as opposed to how to access the different sources of drone law. Nevertheless, the issues raised of legal fragmentation and complexity of the rules are relevant. My specific comments can be found in the text. Overall, the paper is interesting and is going in the right direction, however, I do not consider the analysis to be complete. There are some misunderstandings about EU law, not all potentially relevant laws are mentioned, and key drone law literature does not appear to be consulted.

The English is generally of a high quality, with only some minor issues. Grammerly should solve these.
Author Response

(The authors gave the same response as above.)
